Serum ferritin associated with atherogenic lipid profiles in a high-altitude living general population

Jin Menglong 1
Mamute Mawusumu 2
Shapaermaimaiti Hebali 3
Ji Hongyu 1
Cao Zichen 1
Luo Sifu 1
Abudula Mayire 1
Aigaixi Abuduhalike 4
Fu Zhenyan fuzhenyan316@126.com 1
1 Department of Cardiology, State Key Laboratory of Pathogenesis, Prevention and Treatment of High Incidence Diseases in Central Asia, the First Affiliated Hospital of Xinjiang Medical University , Urumqi , Xinjiang , China
2 Department of Urology, First People’s Hospital of Kashgar District , Kashgar , Xinjiang , China
3 Disease Control and Prevention Center of Tashkurgan Tajik Autonomous County , Tashkurgan , Xinjiang , China
4 Health Commission of Tashkurgan Tajik Autonomous County , Tashkurgan , Xinjiang , China
Foti Daniela
Electronic publication date: 2025 Mar 24
Publication date: 2025
Volume: 13
Electronic Location ID: e19104
Received 2024 Sep 13; Accepted 2025 Feb 12
Copyright: ©2025 Jin et al.
Copyright year: 2025
Copyright holder: Jin et al.
License: This is an open access article distributed under the terms of the Creative Commons Attribution License, which permits unrestricted use, distribution, reproduction and adaptation in any medium and for any purpose provided that it is properly attributed. For attribution, the original author(s), title, publication source (PeerJ) and either DOI or URL of the article must be cited.
License URL: https://creativecommons.org/licenses/by/4.0/

Keywords: High altitude, Atherogenic lipid profiles, SdLDL-C, Lp(a), Serum ferritin

Funding: The Key research and development project of Xinjiang Uygur Autonomous Region 2022B03022-4 “Tianshan Talents” training program 2022TSYCLJ0030 The Science and Technology Support Project of Xinjiang 2024E02043 The National Natural Science Foundation of China 82460097 The Key Project of Xinjiang Natural Science Foundation 2024D01D22 This work was funded by the Key research and development project of Xinjiang Uygur Autonomous Region (2022B03022-4), “Tianshan Talents” training program (2022TSYCLJ0030), the Science and Technology Support Project of Xinjiang (2024E02043), the National Natural Science Foundation of China (82460097), the Key Project of Xinjiang Natural Science Foundation (2024D01D22). The funders had no role in study design, data collection and analysis, decision to publish, or preparation of the manuscript.

==============================
Background

Serum ferritin (SF) levels are associated with metabolic syndrome and dyslipidemia. However, the association between SF and atherogenic lipid profiles in high-altitude living populations remains unclear.

Methods

In 2021, a cross-sectional study was conducted on adult Tajik individuals residing in Tashkurgan Tajik Autonomous County (average altitude 3,100 meters). Demographic information and anthropometric measurements were collected in local clinics. Fasting blood samples were analyzed using a Beckman AU-680 Automatic Biochemical analyzer at the biochemical laboratory of Fuwai Hospital. Univariate linear regression analyses were used to explore the association between SF and atherogenic lipid levels. Subgroup analysis was used based on gender and different high-sensitivity C-reactive protein (hs-CRP) and serum amyloid A (SAA) levels. The association between higher SF quartiles and different kinds of dyslipidemia were analyzed by logistic regression.

Results

There were 1,703 participants in total, among which 866 (50.9%) being men. The mean ages of male and female participants were similar (41.50 vs. 42.38 years; P = 0.224). SF levels were significantly correlated with total cholesterol (TC) (Beta = 0.225, P < 0.001), low-density lipoprotein cholesterol (LDL-C) (Beta = 0.197, P < 0.001), high-density lipoprotein cholesterol (HDL-C) (Beta = −0.218, P < 0.001), triglycerides (TG) (Beta = 0.332, P < 0.001), and small dense LDL-C (sdLDL-C) (Beta = 0.316, P < 0.001), with the exception of lipoprotein (a) (Lp(a)) (Beta = 0.018, P = 0.475). SF was significantly correlated with LDL-C and HDL-C in women, and correlated with TC, TG, and sdLDL-C levels in both men and women in different inflammatory conditions. Elevated SF levels was significantly correlated with high TC (OR: 1.413, 95% CI [1.010–1.978]), high TG (OR: 1.602, 95% CI [1.299–1.976]), and high sdLDL-C (OR: 1.631, 95% CI [1.370–1.942]) in men and high TC (OR: 1.461, 95% CI [1.061–2.014]), high LDL-C (OR: 2.104, 95% CI [1.481–2.990]), low HDL-C (OR: 1.447, 95% CI [1.195–1.752]), high TG (OR: 2.106, 95% CI [1.454–3.050]), and high sdLDL-C (OR: 2.000, 95% CI [1.589–2.516]) in women. After adjusting for potential confounders, elevated SF levels continue to be correlated with high TG in male (OR: 1.382, 95% CI [1.100–1.737]) and female (OR: 1.677, 95% CI [1.070–2.628]) participants. In both young and middle-aged subgroups, the associations between SF and TG, TC, HDL-C, LDL-C, and sdLDL-C were still significant.

Conclusions

SF was closely related to atherogenic lipid profiles, especially with regard to TG in high-altitude populations. This association cannot be attributed to its role as an inflammation marker.

Introduction

Globally, cardiovascular disease (CVD) ranks as the top cause of death (Arnett et al., 2019; Visseren et al., 2021). Low-density lipoprotein cholesterol (LDL-C) is a recognized independent factor contributing to CVD risk and a primary target for CVD prevention (Mach et al., 2020). A subset of LDL, known as small dense LDL (sdLDL), holds a particularly strong association with the risk of atherosclerotic CVD (ASCVD) (Krauss, 2022; Yanai et al., 2022). Notably, lipoprotein a (Lp(a)) has been proven as a causal risk factor for ASCVD, independent of LDL-C (Reyes-Soffer et al., 2022). Additionally, it has been found that triglycerides (TG) and triglyceride-rich lipoproteins (TGRL) play an important in the residual risk realated to ASCVD (Dhindsa et al., 2020).

Inflammation has various maladaptive functions that lead to the progression and instability of ASCVD, and high-sensitivity C-reactive protein (hs-CRP) serve as a main clinical indicator of inflammation (Lawler et al., 2021). Furthermore, serum amyloid A (SAA) and serum ferritin (SF) are also important in acute and chronic inflammation, and are used as additional markers of inflammatory conditions (Kell & Pretorius, 2014; Zhang et al., 2019).

SF, apart from being an inflammation marker, is also vital in the iron delivery system and is gauged as an indicator of iron status (Kell & Pretorius, 2014; Lopez et al., 2016). People who live at high altitudes for a long time are constantly in a hypoxic environment. Erythrocytosis is an important adaptation mechanism that requires substantially higher amounts of iron (Gassmann & Muckenthaler, 2015). A study on young Swiss male whose residential altitudes range from 200 to 2,000 m revealed that for every 300 m increase in residential altitude, the levels of hemoglobin (Hb) and SF increased significantly (Staub et al., 2020). However, a study conducted in Han Chinese males showed that SF levels of participants living at high altitudes did not increase compared with those of sea level residents (Liu et al., 2018). The iron stores of high-altitude residents were distinct among different populations. In healthy people, iron stores were in the physiological range in spite of the increased need for erythropoiesis. However, for vulnerable groups living at a high altitude with a greater demand for iron, their iron stores were not likely to be quickly replenished as in normal situations (Böning et al., 2004; Burke et al., 2017; Cook et al., 2005; Tufts et al., 1985).

Preliminary studies conducted in different populations, including Spanish (González et al., 2006), American (Li et al., 2023), Qatar (Al Akl et al., 2021), and Chinese (Li et al., 2017; Zhou, Liu & Yuan, 2021) adults as well as Korean (Kim et al., 2016) and Chinese (Zhu et al., 2019) adolescents, have demonstrated that SF was significantly associated with dyslipidemia. Nevertheless, the association between SF and atherogenic lipid profiles in high-altitude living populations was unclear. Given the remarkable effect of high altitude on iron stores and SF levels, studies investigating the correlations between SF and dyslipidemia specifically in high-altitude residents are necessary. To fill this knowledge gap, we investigate for the first time the association between SF and atherogenic lipid profiles in a general population of Tajik adults living on the Pamirs Plateau of China.

Materials & Methods

Study population

In 2021, we conducted a cross-sectional study in northwestern Xinjiang, China. The eligible participants were Tajik individuals aged 18 years or older residing in Tashkurgan Tajik Autonomous County, which boasts an average altitude of 3,100 m. A total of 1,830 individuals were initially recruited for this study, with 1,753 completing the survey. Subsequently, 50 participants with a history of stroke, chronic inflammatory diseases, coronary heart disease (CHD), or those currently on lipid-lowering drugs were excluded. Ultimately, 1,703 participants were incorporated into the analysis. All participants provided written informed consent, and the study was approved by the Ethics Committee of the First Affiliated Hospital of Xinjiang Medical University (210521 - 02).

Data collection

Sociodemographic information (age, gender, educational background, smoking habits, alcohol consumption, and disease history) was collected through questionnaires. After a 5-minute rest, two blood pressure (BP) measurements were made. The means of these two readings was employed for analysis. We measured waist circumference (WC), height, and body weight when participants were in lightweight clothing and without shoes. We calculated body mass index (BMI) as follows: BMI (kg/m2) = weight (kg)/(height (m))2. Blood samples were collected after an 8–14 h overnight fast as recommended by the World Health Organization (Alberti & Zimmet, 1998). These blood samples were then centrifuged at 3,000 rpm for 5 min, after which the serums were rapidly frozen at −80 degrees Celsius. Subsequently, the samples were transported to the biochemical laboratory of the Department of Epidemiology of Fuwai Hospital. This laboratory was part of the Lipid Standardization Program run by the US Centers for Disease Control and Prevention, ensuring the accuracy of tests conducted. Fast blood glucose (FBG), TC, TG, high-density lipoprotein cholesterol (HDL-C), Lp(a), sdLDL-C, ferritin, hs-CRP, and SAA were tested using a Beckman AU-680 Automatic Biochemical analyzer. The Friedewald formula was used to calculate LDL-C. LDL-C (mmol/L) = TC (mmol/L)−HDL-C (mmol/L)−TG (mmol/L)/2.2. The collection of questionnaires, physical examinations, and blood samples was carried out by trained staff at the local health center.

Outcome definition

We used the 2023 Chinese guidelines for the management of dyslipidemia in adults to define different types of dyslipidemia (Joint Committee on the Chinese Guidelines for Lipid Management, 2023). High TC, high LDL-C, high TG, low HDL-C, high Lp(a), and high sdLDL-C were defined as TC ≥ 6.22 mmol/L, LDL-C ≥ 4.14 mmol/L, TG ≥ 2.26 mmol/L, HDL-C < 1.04 mmol/L, Lp(a) > 300 mg/L, and sdLDL-C ≥ 1.4 mmol/L, respectively. Smoking was defined as smoking at least 100 cigarettes and smoking within the last month. Drinking was defined as drinking at least 12 times in the last year. Exercise was defined as engaging in moderate or vigorous physical activity ≥ 150 min per week.

Statistical analyses

IBM SPSS Statistics 26 was used to do statistical analyses. Continuous variables were shown as means ± standard deviation (SD) (normal distribution) or as quartiles (non-normal distribution). Category variables were represented as numbers (percentages). T-tests were utilized to evaluate the differences in continuous variables that followed a normal distribution, while Mann–Whitney U tests were employed for those that did not adhere to a normal distribution. Chi-square tests were utilized to investigate differences among category variables. Standardized regression coefficients (Beta) calculated by univariate linear regression analyses were used to assess the correlation between SF levels and lipids. Univariate logistic regression analyses were used to test the trend of dyslipidemia along ferritin. Multivariate logistics regression analyses were used to exclude the influence of confounder on dyslipidemia. P < 0.05 was considered statistically significant.

Results

Population characteristics

Among the 1,703 participants, 866 (50.9%) were men. The average ages of men and women were similar (41.50 vs 42.38 years; P = 0.224). The mean hemoglobin levels of Tajik reached up to 158.88g/L with a significantly gender discrepancy (168.17 vs 149.28 g/l; P < 0.001). Male participants had larger WC (mean 80.83 vs 76.16 cm; P < 0.001), slower heart rates (mean 65.11 vs 67.94 times/minute; P = 0.001), and higher rates of smoking (36.3% vs 0.1%; P < 0.001) and drinking (14.5% vs 0.1%; P < 0.001) than females, but the age, BMI, BP, and prevalence of hypertension and diabetes were similar. Regarding inflammation markers, men had lower SAA levels (median 5.30 vs 5.50 mg/L; P < 0.001) and notably higher SF levels (median 200.20 vs 42.30 ng/mL; P < 0.001) than women. There was no significant difference in hs-CRP concentration between men and women (median 4.28 vs 4.58 mg/L; P = 0.098). In terms of lipid profiles, men had higher TG (median 1.05 vs 0.84 mmol/L; P < 0.001), ApoB (mean 0.75 vs 0.72 g/L; P = 0.001), and sdLDL-C (mean 37.63 vs 32.85 mmol/L; P < 0.001) concentration but lower HDL-C (mean 1.18 vs 1.37; P < 0.001) and ApoA1 (mean 1.19 vs 1.29 g/L; P < 0.001) concentration than women (Table 1).

Table 1 Demographics and baseline characteristics of participants.

	Total
(n = 1703)	Men
(n = 866)	Women
(n = 837)	P value	
Age (years)	41.93 ± 14.83	41.50 ± 14.78	42.38 ± 14.89	0.224	
Hypertension N (%)	359(21.1)	170(19.6)	189(22.6)	0.136	
Diabetes N (%)	33(1.9)	22(2.6)	11(1.3)	0.066	
Smoking N (%)	307(18.0)	306(36.3)	1(0.1)	<0.001	
Drinking N (%)	123(7.2)	122(14.5)	1(0.1)	<0.001	
Exercise N (%)	216(12.7)	160(19.0)	56(6.7)	0.079	
WC (cm)	78.54 ± 14.74	80.83 ± 15.48	76.16 ± 13.54	<0.001	
BMI (kg/cm2)	22.25 ± 12.69	21.91 ± 3.16	22.60 ± 17.81	0.266	
SBP (mmHg)	120.18 ± 24.84	119.82 ± 23.12	120.56 ± 26.52	0.542	
DBP (mmHg)	74.70 ± 15.90	74.55 ± 15.16	74.85 ± 16.64	0.701	
HR (times/minute)	66.50 ± 16.86	65.11 ± 15.83	67.94 ± 17.77	0.001	
FBG (mmol/L)	4.89 ± 1.19	4.91 ± 1.01	4.86 ± 1.35	0.386	
hs-CRP (mg/L)	4.41 (2.71, 5.37)	4.28 (2.58, 5.32)	4.58 (2.83, 5.44)	0.098	
SAA (mg/L)	5.40 (4.80, 6.70)	5.30 (4.80, 6.30)	5.50 (4.90, 7.10)	<0.001	
Ferritin (ng/mL)	127.90 (37.15, 227.20)	200.20 (134.20, 304.20)	42.30 (12.30, 118.50)	<0.001	
TC (mmol/L)	4.46 ± 0.92	4.45 ± 0.92	4.48 ± 0.92	0.484	
LDL-C (mmol/L)	2.65 ± 0.79	2.65 ± 0.79	2.64 ± 0.79	0.775	
HDL-C (mmol/L)	1.28 ± 0.34	1.18 ± 0.34	1.37 ± 0.31	<0.001	
TG (mmol/L)	0.93 (0.69, 1.37)	1.05 (0.78, 1.55)	0.84 (0.64, 1.17)	<0.001	
ApoA1 (g/L)	1.24 ± 0.23	1.19 ± 0.23	1.29 ± 0.22	<0.001	
ApoB (g/L)	0.74 ± 0.17	0.75 ± 0.17	0.72 ± 0.17	0.001	
Lp(a) (mg/L)	86.00 (43.85, 200.05)	89.80 (44.80, 209.15)	81.35 (43.00, 185.60)	0.321	
sdLDL-C (mg/L)	35.28 ± 15.84	37.63 ± 15.67	32.85 ± 15.65	<0.001	
Notes.

Abbreviations N number

WC waist circumference

BMI body mass index

SBP systolic blood pressure

DBP dilation blood pressure

FBG fast blood glucose

HR heart rate

hs-CRP high-sensitivity C-reactive protein

SAA serum amyloid A

TC total cholesterol

LDL-C low-density lipoprotein cholesterol

HDL-C high-density lipoprotein cholesterol

TG triglycerides

ApoA1 apolipoprotein A1

ApoB apolipoprotein B

Lp(a) lipoprotein a

sdLDL-C small dense LDL-C

Continuous variables were shown as means ± standard deviation.

Categorical variables were shown as numbers (percentages).

Correlation between SF and atherogenic lipid profiles

SF was significantly correlated with most of the lipids, including TC (Beta = 0.225, P < 0.001), LDL-C (Beta = 0.197, P < 0.001), HDL-C (Beta = −0.218, P < 0.001), TG (Beta = 0.332, P < 0.001), and sdLDL-C (Beta = 0.316, P < 0.001). An exception was observed with Lp(a), which showed no significant correlation (Beta = 0.018, P = 0.475) (Fig. 1).

Figure 1 Correlative analyses were conducted between serum ferritin (ng/ml) and (A) TC (mmol/L), (B) LDL-C (mmol/L), (C) HDL-C (mmol/L), (D) TG (mmol/L), (E) Lp(a) (mg/L), and (F) sdLDL-C (mg/L).

TC, total cholesterol; LDL-C, low-density lipoprotein cholesterol; HDL-C, high-density lipoprotein cholesterol; TG, triglycerides; Lp(a), lipoprotein a; sdLDL-C, small and dense LDL-C.

Sex-specific correlation between SF concentration and atherogenic lipids in different inflammatory states

SF was not significantly correlated with hs-CRP (r = 0.015, P = 0.537) and SAA (r = 0.004, P = 0.854), but there was a significant correlation between hs-CRP and SAA (r = 0.753, P < 0.001). To ascertain whether the correlation between SF and lipid profiles remained independent of the inflammatory states, subgroup analyses were conducted, stratifying the participants based on the interquartile range of hs-CRP and SAA levels. Given the significant difference in SF between men and women, sex-specific correlation analyses were performed. SF exhibited significant correlations with TC, TG, and sdLDL-C levels in all subgroups, in both men and women. In women, SF was significantly correlated with LDL-C and HDL-C in all subgroups. No significant correlation was observed between SF concentration and Lp(a) levels in any of the subgroups (Table 2).

Association of SF concentration with different types of dyslipidemia

In the male population, higher SF was positively correlated with high TC (OR: 1.413, 95% CI [1.010–1.978]), high TG (OR: 1.602, 95% CI [1.299–1.976]), and high sdLDL-C (OR: 1.631, 95% CI [1.370–1.942]). After adjusting for age (model 1), SF maintained a significant correlation with high TG (OR: 1.610, 95% CI [1.303–1.990]) and high sdLDL-C (OR: 1.610, 95% CI [1.353–1.917]). In model 2 (adjusting for WC, BMI, and BP based on model 1) and model 3 (adjusting for smoking, drinking, and exercise based on model 2), SF concentration remained significantly correlated with high sdLDL-C (OR: 1.470, 95% CI [1.224–1.766]) and high TG (OR: 1.382, 95% CI [1.100–1.737]) (Fig. 2).

Figure 2 Odds ratios (ORs) and their 95% confidence intervals (95% CIs) for (A) high TC, (B) high LDL-C, (C) low HDL-C, (D) high TG, (E) high Lp(a), and (F) high sdLDL-C according to serum ferritin quartile in males.

TC, total cholesterol; LDL-C, low-density lipoprotein cholesterol; HDL-C, high-density lipoprotein cholesterol; TG, triglycerides; Lp(a), lipoprotein a; sdLDL-C, small and dense LDL-C; Crude, ORs with no adjustments. Model 1: ORs adjusted for age. Model 2: ORs adjusted for waist circumference, BMI, and blood pressure based on model 1. Model 3: ORs adjusted for smoking, drinking and exercise based on model 2. Definitions of different types of dyslipidemia are described in the Methods section.

In the female population, higher SF was positively correlated with high TC (OR: 1.461, 95% CI [1.061–2.014]), high LDL-C (OR: 2.104, 95% CI [1.481–2.990]), low HDL-C (OR: 1.447, 95% CI [1.195–1.752]), high TG (OR: 2.106, 95% CI [1.454–3.050]), and high sdLDL-C (OR: 2.000, 95% CI [1.589–2.516]). In model 1, higher SF was significantly correlated with high TG (OR: 1.970, 95% CI [1.260–3.079]), low HDL-C (OR: 1.416, 95% CI [1.120–1.790]), and high sdLDL-C (OR: 1.424, 95% CI [1.087–1.865]). In model 2 and model 3, higher SF concentration remained significantly correlated with low HDL-C (OR: 1.348, 95% CI [1.061–1.711]) and high TG (OR: 1.677, 95% CI [1.070–2.628]) (Fig. 3).

Table 2 Sex specific correlations between serum ferritin concentrations and lipid profiles stratified by hs-CRP and SAA levels.

	TC (mmol/L)	LDL-C (mmol/L)	HDL-C (mmol/L)	TG (mmol/L)	Lp(a) (mg/L)	sdLDL-C (mg/L)	
	Beta	P	Beta	P	Beta	P	Beta	P	Beta	P	Beta	P	
Men													
hs-CRP (mg/L)													
Q1	0.245	<0.001	0.066	0.333	−0.055	0.427	0.334	<0.001	0.007	0.922	0.154	0.024	
Q2	0.290	<0.001	0.231	0.001	0.004	0.955	0.168	0.013	−0.057	0.405	0.309	<0.001	
Q3	0.139	0.044	0.052	0.454	0.060	0.383	0.185	0.007	−0.061	0.381	0.177	0.010	
Q4	0.186	0.005	0.155	0.021	−0.069	0.306	0.222	0.001	0.134	0.050	0.231	0.001	
SAA (mg/L)													
Q1	0.295	<0.001	0.250	<0.001	−0.003	0.965	0.212	0.003	0.025	0.725	0.289	<0.001	
Q2	0.133	0.045	0.089	0.180	−0.087	0.192	0.202	0.002	0.004	0.954	0.201	0.002	
Q3	0.194	0.005	−0.011	0.875	−0.110	0.110	0.375	<0.001	−0.110	0.114	0.151	0.028	
Q4	0.249	<0.001	0.176	0.008	0.078	0.243	0.164	0.013	0.062	0.361	0.241	<0.001	
Women													
hs-CRP (mg/L)													
Q1	0.324	<0.001	0.349	<0.001	−0.254	<0.001	0.361	<0.001	−0.072	0.307	0.369	<0.001	
Q2	0.307	<0.001	0.327	<0.001	−0.193	0.005	0.275	<0.001	0.091	0.193	0.297	<0.001	
Q3	0.373	<0.001	0.409	<0.001	−0.216	0.002	0.275	<0.001	0.024	0.729	0.414	<0.001	
Q4	0.279	<0.001	0.313	<0.001	−0.219	0.001	0.337	<0.001	−0.043	0.546	0.341	<0.001	
SAA (mg/L)													
Q1	0.359	<0.001	0.355	<0.001	−0.247	0.001	0.381	<0.001	−0.072	0.362	0.453	<0.001	
Q2	0.375	<0.001	0.390	<0.001	−0.288	<0.001	0.376	<0.001	0.115	0.087	0.410	<0.001	
Q3	0.340	<0.001	0.369	<0.001	−0.201	0.002	0.299	<0.001	−0.006	0.930	0.329	<0.001	
Q4	0.188	0.006	0.263	<0.001	−0.275	<0.001	0.248	<0.001	−0.035	0.618	0.271	<0.001	
Notes.

hs-CRP high-sensitivity C-reactive protein

SAA serum amyloid A

TC total cholesterol

LDL-C low-density lipoprotein cholesterol

HDL-C high-density lipoprotein cholesterol

TG triglycerides

Lp(a) lipoprotein a

sdLDL-C small dense LDL-C

Q1 first quartile

Q2 second quartile

Q3 third quartile

Q4 fourth quartile

Beta standardized regression coefficients

P p values

Figure 3 Odds ratios (ORs) and their 95% confidence intervals (95% CIs) for (A) high TC, (B) high LDL-C, (C) low HDL-C, (D) high TG, (E) high Lp(a), and (F) high sdLDL-C according to serum ferritin quartile in females.

TC, total cholesterol. lipoprotein cholesterol; HDL-C, high-density lipoprotein cholesterol; TG, triglycerides; Lp(a), lipoprotein a; sdLDL-C, small and dense LDL-C; Crude, ORs with no adjustments. Model 1: ORs adjusted for age. Model 2: ORs adjusted for waist circumference, BMI, and blood pressure based on model 1. Model 3: ORs adjusted for smoking, drinking and exercise based on model 2. Definitions of different types of dyslipidemia are described in the Methods section.

Subgroup analysis for young (18–35 years) and middle-aged (≥45 years) participants

In the young participants group, SF was significantly correlated with TC (Beta = 0.192, P < 0.001), LDL-C (Beta = 0.168, P < 0.001), HDL-C (Beta = −0.313, P < 0.001), TG (Beta = 0.374, P < 0.001), and sdLDL-C (Beta = 0.366, P < 0.001). There was no correlation between SF and Lp(a) (Beta = −0.008, P = 0.845) (Fig. S1). In the middle-aged participants group, SF was significantly correlated with TC (Beta = 0.162, P < 0.001), LDL-C (Beta = 0.110, P = 0.005), HDL-C (Beta = −0.162, P < 0.001), TG (Beta = 0.299, P < 0.001), and sdLDL-C (Beta = 0.214, P < 0.001). There was no correlation between SF and Lp(a) (Beta = −0.001, P = 0.977) (Fig. S2).

In the young population, higher SF quartile was positively correlated with high TG (OR: 1.702, 95% CI [1.203–2.407]) and high sdLDL-C (OR: 1.434, 95% CI [1.061–1.937]). There was no correlation between SF quartile and high TC (OR: 1.167, 95% CI [0.677–2.011]), high LDL-C (OR: 0.693, 95% CI [0.295–1.624]), low HDL-C (OR: 1.123 95% CI [0.930–1.356]), and high Lp(a) (OR: 0.927, 95% CI [0.724–1.167]) after adjusting for gender, WC, BMI, SBP, smoking, drinking, and exercise (Fig. 4A). In the middle-aged participants group, higher SF quartile was positively correlated with high LDL-C (OR: 1.459, 95% CI [1.042–2.044]), high TG (OR: 1.325, 95% CI [1.003–1.749]), and high sdLDL-C (OR: 1.502, 95% CI [1.218–1.852]). There was no correlation between SF quartile and high TC (OR: 1.307, 95% CI [0.926–1.844]), low HDL-C (OR: 1.016 95% CI [0.853–1.209]), or high Lp(a) (OR: 1.007, 95% CI [0.891–1.302]) (Fig. 4B).

Figure 4 Subgroup analysis of the odds ratios (ORs) of different types of dyslipidemia according to serum ferritin quartiles for young (18-35 years) (A) and middle-aged (≥ 45 years) (B) participants.

High TC, high total cholesterol; High LDL-C, high low-density lipoprotein cholesterol; Low HDL-C, low high-density lipoprotein cholesterol. High TG, high triglycerides; High Lp(a), high lipoprotein a. High sdLDL-C, small and dense LDL-C. Definitions of different types of dyslipidemia are described in the Methods section.

Discussion

In this study, it was found that SF concentration was significantly higher in men compared to women. SF exhibited positively correlations with TC, LDL-C, TG, and sdLDL-C. Conversely, it displayed a negatively correlation with HDL-C. There was no significant correlation with Lp(a). Under different inflammatory conditions, SF was positively correlated with LDL-C and negatively correlated with HDL-C in women. SF was positively correlated with TC, TG, and sdLDL-C in both men and women. Higher SF levels were associated with high TC, high TG, and high sdLDL-C in men, and almost all kinds of dyslipidemia except for high Lp(a) in women. Even after adjusting for confounders, higher SF levels remained significantly associated with high TG in both men and women. The correlations between SF and TC, LDL-C, HDL-C, TG, and sdLDL-C were still significant in both young and middle-aged subgroups.

People who have long-term residence at high altitudes are in a hypoxic environment. Hypoxemia is sensed by prolyl hydroxylases (PHDs), which stabilize hypoxia-inducible factor (HIF). HIF can promote the expression of erythropoietin (Epo), increasing red blood cell production (Gassmann & Muckenthaler, 2015). Epo, hepcidin, and erythroferrone are involved in coordinating the liver, bone marrow, and small intestine, as well as altering iron acquisition. Therefore, altitude is closely related to iron metabolism. In this study, the mean Hb levels of high-altitude Tajik were up to 168.17 g/L in males and 149.28 g/l in females, indicating a high-altitude adaptation. Several studies conducted in different populations living at high altitudes showed that iron stores were distinct in healthy individuals and vulnerable populations with increased iron demand (Böning et al., 2004; Burke et al., 2017; Cook et al., 2005; Tufts et al., 1985). We reported that the median SF of Tajik males and females were 200.20 ng/mL and 42.30 ng/mL, respectively. Compared to the study conducted on Bolivian adults living at 3,500 m of altitude or above, the SF levels of male Tajik were much higher, but the SF levels of female Tajiks were similar (Beall et al., 1990). Regrettably, we have no data on SF in low-altitude Tajik populations since most Tajiks live on the high-altitude Pamirs Plateau of China, which prevented us from investigating the effects of high altitude on iron store.

Previous research indicated a consistent pattern regarding the relationship between SF and lipid profiles, showing positive correlations with TC, LDL-C, and TG, while revealing negative correlation with HDL-C levels in populations from the United States (Li et al., 2023) and Middle East (Al Akl et al., 2021). In this study conducted among high-altitude Tajik individuals, these trends were similarly observed. Notably, serum Lp(a) levels are mainly determined by genetic factors (Kronenberg et al., 2022; Reyes-Soffer et al., 2022), and the prevalence of high Lp(a) remained unassociated with SF in both men and women in this study. The exact way in which SF concentrations are related to dyslipidemia has not been fully grasped yet. Numerous studies have proven that SF was associated with other CVD risk factors, such as diabetes (Sun et al., 2013), hypertension (Piperno et al., 2002), elevated fasting insulin and blood glucose (Tuomainen et al., 1997), central adiposity (Gillum, 2001), and metabolic syndrome (Jehn, Clark & Guallar, 2004; Suárez-Ortegón et al., 2018), suggesting that SF may play a significant role in metabolism. SF could lose its iron and the unliganded iron has the capacity to catalyze hydroxyl radical formation, leading to oxidative tissue damage (Kell & Pretorius, 2014; Wolff, 1993). Studies on rats showed that dietary iron restriction can lead to decreased TG concentrations and reduced levels of lipid peroxidation levels (Vargas-Vargas et al., 2022). However, iron overload may induce elevated TG levels through several mechanisms, including enhanced hepatic oxidative stress, decreased fatty acid beta-oxidation, and the promoted hepatic lipid secretion by increasing the expression of apoB-100 and microsomal triglyceride transfer protein (Silva et al., 2015).

SF, apart from being a sign of iron storage, serves as a crucial marker for inflammatory diseases too, since it is mainly a product leaked from damaged cells (Kell & Pretorius, 2014). Therefore, is the relationship between SF and lipid profiles in high-altitude residents associated with its role as an inflammation marker? A previous study conducted in the general Chinese population found that the association between SF and lipid parameters persisted after adjusting for CRP levels, suggesting that low-grade inflammation may not be the sole explanation for the link between SF levels and dyslipidemia (Li et al., 2017). In this study, individuals with chronic inflammatory diseases were excluded. Notably, SF showed no significant correlation with two other inflammation markers, hsCRP and SAA. Moreover, SF remained significantly associated with TC, TG, and sdLDL-C across subgroups stratified by varying levels of hs-CRP and SAA. These findings suggest that the relationship between SF and lipid levels cannot be attributed to its role as an inflammation marker.

There seemed to be a gender discrepancy in the association between SF and lipids levels or dyslipidemia. A study conducted on Korean adolescents found that SF was positively associated with TC, LDL-C, and TG levels, and negatively associated with HDL-C levels in women. However, in men, the association was observed only with HDL-C levels (Kim et al., 2016). This result differed from a study conducted in US adults, where only females showed a clearly increasing tendency for high-TC and high-LDL-C. In this study, the relationship between SF and lipid levels appeared to be stronger in women, which aligned with the findings from the US study. In women, five out of six lipid parameters were correlated with SF across varying levels of inflammation. In contrast, only three out of six lipid parameters were significantly associated with SF in men. Regarding dyslipidemia, nearly all types of dyslipidemia, except for high Lp(a), were associated with higher SF in women. However, after adjusting for confounders, SF was only associated with high TG in both men and women, indicating that the gender discrepancy in the association between SF and lipids might be attributed to differences in the distributions of confounders between men and women.

There are a number of limitations in this study which need to be taken into account. First, because of its cross-sectional nature, the study was unable to determine a causal connection between SF and lipid profiles. Prospective cohort studies would be necessary to determine a causal relationship. Second, although this study was conducted in a single ethnic group with similar eating habits, dietary information was not included in the study, making it challenging to adjust for dietary factors (especially iron intake) in the analysis of the relationship between SF and lipids. Third, the menstrual status of women was unknown, which might have introduced potential bias to the results.

Despite these limitations, a key strength of this study lies in its novel investigation of the association between SF and lipid profiles under varying inflammatory conditions in high-altitude living residents. Moreover, as far as we know, this is the first study to look into the relation between SF and Lp(a) as well as sdLDL-C. These findings help us better understand the interplay between SF and lipids, providing useful perspectives and create a basis for more researches in the future.

In conclusion, SF was positively correlated with TC, LDL-C, TG, and sdLDL-C; negatively correlated with HDL-C; and not significantly correlated with Lp(a). Even after adjusting for confounders, higher SF levels were associated with high TG in both genders. The relationship between SF and lipid levels cannot be attributed to its role as an inflammation marker.

Supplemental Information

Supplemental Information 1 STROBE checklist

Supplemental Information 2 Correlative analyses were conducted between serum ferritin (ng/ml) and A: TC (mmol/L), B: LDL-C (mmol/L), C: HDL-C (mmol/L), D: TG (mmol/L), E: Lp(a) (mg/L), and F: sdLDL-C (mg/L) in young (18-35 years) participants

TC: total cholesterol. LDL-C: low-density lipoprotein cholesterol. HDL-C: high-density lipoprotein cholesterol. TG: triglycerides. Lp(a): lipoprotein a. sdLDL-C: small and dense LDL-C.

Supplemental Information 3 Correlative analyses were conducted between serum ferritin (ng/ml) and A: TC (mmol/L), B: LDL-C (mmol/L), C: HDL-C (mmol/L), D: TG (mmol/L), E: Lp(a) (mg/L), and F: sdLDL-C (mg/L) in middle-aged (≥ 45 years) participants

TC: total cholesterol. LDL-C: low-density lipoprotein cholesterol. HDL-C: high-density lipoprotein cholesterol. TG: triglycerides. Lp(a): lipoprotein a. sdLDL-C: small and dense LDL-C.

Supplemental Information 4 Raw data

We acknowledge the support provided by the People‘s Hospital of Tashkurgan Tajik Autonomous County and Tashkurgan government. We acknowledge the contribution made by all staff members and the participation of all participants.

Additional Information and Declarations

Competing Interests

Author Contributions

Human Ethics

Data Availability

The authors declare there are no competing interests.

Menglong Jin performed the experiments, analyzed the data, prepared figures and/or tables, authored or reviewed drafts of the article, and approved the final draft.

Mawusumu Mamute performed the experiments, prepared figures and/or tables, and approved the final draft.

Hebali Shapaermaimaiti performed the experiments, prepared figures and/or tables, and approved the final draft.

Hongyu Ji performed the experiments, prepared figures and/or tables, and approved the final draft.

Zichen Cao performed the experiments, prepared figures and/or tables, and approved the final draft.

Sifu Luo performed the experiments, authored or reviewed drafts of the article, and approved the final draft.

Mayire Abudula performed the experiments, authored or reviewed drafts of the article, and approved the final draft.

Abuduhalike Aigaixi performed the experiments, prepared figures and/or tables, and approved the final draft.

Zhenyan Fu conceived and designed the experiments, authored or reviewed drafts of the article, and approved the final draft.

The following information was supplied relating to ethical approvals (i.e., approving body and any reference numbers):

The study received approval from the Ethics Committee of the First Affiliated Hospital of Xinjiang Medical University (Approval Number: 210521-02).

The following information was supplied regarding data availability:

The raw measurements are available in the Supplementary Files.

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
