# Peer review of "Serum ferritin associated with atherogenic lipid profiles in a high-altitude living general population"

_PeerJ, doi:10.7717/peerj.19104_

## Round 0.1 · original submission · Major Revisions

To improve the manuscript, please address the concerns raised by both reviewers.

Reviewer 1 ·

Basic reporting

1_The Plagiarism of the article is 35 %, which is very high

2_ I don't understand why the "Methods" section in the abstract is so long.

3_ Unclear, unambiguous, non-professional English language used throughout.

4_Intro show context. Literature well referenced & relevant. But the background could have been better.

5_Structure is to be improved to conform to PeerJ standards and discipline norms or enhanced for clarity.

6_Figures are relevant, but of low quality, well-labelled & not well-described.

Experimental design

1_ The research question is well-defined and relevant, but I feel it can be made more meaningful.

2_ It is not fully stated how the research fills an identified knowledge gap.


3_ Rigorous investigation is not performed to a high technical & ethical standard.

4_Methods described with sufficient detail & information to replicate - but I feel some more references like from WHO in case of fasting (minimum is 8 hrs, but the maximum is?), etc could have made it better

Validity of the findings

1_Impact and novelty are not assessed - I feel they should explain it clearly


2_ Conclusions are okay, linked to the original research question & limited to supporting results. But I feel can be improved

Additional comments

The manuscript should be rigorously revised, plagiarism reduced, and the English language should be more professional

Annotated reviews are not available for download in order to protect the identity of reviewers who chose to remain anonymous.

Reviewer 2 ·

Basic reporting

The manuscript reports association between serum inflammatory markers especially ferritin with lipid profiles in high altitude dwelling Tajik individuals. Notably, the study reports existence of gender differences for serum ferritin levels and the higher levels of serum ferritin is associated with high TG levels in both the genders. This is an interesting study reporting association between inflammation and atherogenic state for a high altitude population.

Experimental design

The research question is well defined supported with appropriate statistical analysis. However, the manuscript can be improved by considering following suggestions:

1) High altitude exposure stimulates erythropoiesis and high altitude populations maintain their iron stores within the physiological range despite elevated requirements for erythropoiesis (Am J Hum Biol 2: 639 –651, 1990; High Alt Med Biol 11: 199 –208, 2010; Blood 135: 1066 –1069, 2020). It is well reported that Indigenous high altitude populations have high levels of serum ferritin (PMID: 34508740) that helps to offset ill effects of hypobaric hypoxia (PMID: 37134197). In this context, it is important to measure and analyze the hemoglobin levels of the studied individuals.

2) In continuation with point 1, several other researchers have reported higher serum ferritin levels at moderate and high altitudes as well as gender differences for serum ferritin levels (J Appl Physiol 129: 920 –925, 2020). The authors may go though this well compiled review paper and discuss their findings accordingly.

3) Is serum ferritin an inflammatory marker at high altitude? The authors need to justify this paradox. The authors may consider report by Kell and Pretorius in this regard (PMID: 24549403).

4) The authors are advised to perform another set of analysis with young participants (age group 18 -35) and middle aged participants ( age 45 and above) for assessing atherogenic lipid profile.

5) 4) The authors have excluded participants with lipid lowering medications in the present study. Observations from the present study may be validated in such participants.

Validity of the findings

The manuscript reports statistical association of serum ferritin with lipid profiles. Since high altitude exposure is a major environmental stress and indigenous high altitude dwellers have developed metabolic adaptations to survive in these challenging conditions, the authors are advised to consider and incorporate high altitude-induced ferritin as well as lipid profiles for better analysis of the observed results.

Additional comments

NIL

---

## Round 0.2 · accepted · Accept

The authors have satisfactorily addressed all the concerns raised, so that the study can be published in PeerJ in the present form.

Reviewer 2 ·

Basic reporting

The authors have improved the manuscript as per reviewer suggestions. The manuscript can now be accepted for publication.

Experimental design

The authors have performed additional analysis as per reviewer suggestions.

Validity of the findings

The results now can be published in PeerJ.

Additional comments

NIL